# Mitochondrial Neurodegeneration: Lessons from *Drosophila melanogaster* Models

**DOI:** 10.3390/biom13020378

**Published:** 2023-02-16

**Authors:** Michele Brischigliaro, Erika Fernandez-Vizarra, Carlo Viscomi

**Affiliations:** 1Department of Biomedical Sciences, University of Padova, 35131 Padova, Italy; 2Veneto Institute of Molecular Medicine, 35129 Padova, Italy; 3Centre for the Study of Neurodegeneration (CESNE), University of Padova, 35131 Padova, Italy

**Keywords:** mitochondrial disease, neurodegeneration, OXPHOS, *Drosophila melanogaster*

## Abstract

The fruit fly—i.e., *Drosophila melanogaster*—has proven to be a very useful model for the understanding of basic physiological processes, such as development or ageing. The availability of straightforward genetic tools that can be used to produce engineered individuals makes this model extremely interesting for the understanding of the mechanisms underlying genetic diseases in physiological models. Mitochondrial diseases are a group of yet-incurable genetic disorders characterized by the malfunction of the oxidative phosphorylation system (OXPHOS), which is the highly conserved energy transformation system present in mitochondria. The generation of *D. melanogaster* models of mitochondrial disease started relatively recently but has already provided relevant information about the molecular mechanisms and pathological consequences of mitochondrial dysfunction. Here, we provide an overview of such models and highlight the relevance of *D. melanogaster* as a model to study mitochondrial disorders.

## 1. *Drosophila melanogaster* as a Model Organism to Study Disease

The fruit fly (*Drosophila melanogaster*) has been widely used as a model for research in different fields of biology. The main advantages are its short life cycle (Figure 1); small size; rapid reproductive rate, which is useful for empowering statistical analyses; and the possibility of easy and cheap maintenance of many strains in a limited space.

These features make *Drosophila melanogaster* an attractive organism for both basic and applied genetics studies. As approximately 75% of human disease-related genes have a functional homolog in the fruit fly genome [1,2], it also constitutes a good model for the study of human disorders. A particular advantage of using *D. melanogaster* as a model system is access to powerful genetic tools (Figure 2).

For example, transposable P-elements and chemical/physical mutagenesis have been used to induce a large number of mutations and deletions [3,4]. P-elements are transposons that have been engineered to induce genetic modifications through insertional mutagenesis. In addition, the heterologous UAS/GAL4 dual system from *S. cerevisiae* has been transferred to *D. melanogaster* in order to finely control the spatiotemporal expression and knockdown of any gene [5,6,7]. This makes it possible to mimic hypomorphic mutations and to overcome frequent experimental limitations linked to lethal or severe phenotypes associated with the complete genetic knockout of essential genes. In fact, the GAL4 transcriptional activator can be expressed under the control of tissue-specific or stage-specific promoters, inducing the expression of elements such as transgenes or inverted-repeats (i.e., single-stranded sequences of nucleotides followed downstream by their reverse complement) for gene knockdown, located downstream of the UAS. Moreover, tissue-specific manipulations are valuable tools that can be used to determine the contribution of each tissue to a disease phenotype. Finally, precise genome-editing techniques developed in the last decade, such as TALENs [8], the CRISPR/Cas9 system [9,10,11], and, more recently, base editing and prime editing [12,13], have been successfully applied to modify the *Drosophila* genome [14].

Thus, the use of *D. melanogaster* models provides a rather large array of genetic tools for the investigation of the molecular pathogenesis of human diseases.

## 2. Mitochondrial Diseases

Mitochondria are double-membrane organelles responsible for the production of most of the ATP in cells via the process of oxidative phosphorylation (OXPHOS) (Figure 3).

The OXPHOS system is composed of four respiratory complexes (complexes I–IV) and two electron carriers (coenzyme Q and cytochrome *c*), through which an electron funneling cascade coupled with proton pumping allows the generation of an electrochemical gradient across the inner mitochondrial membrane. The electrochemical gradient generates a proton-motif force (pmf) that is exploited by a fifth complex (complex V, ATP synthase) to synthesize ATP from ADP and inorganic phosphate (P_i_). ATP stores energy within its phosphodiester bonds, which is then released through the hydrolysis of the bond between the b and g phosphates, driving practically all endergonic biological processes. Although ATP synthesis is often considered the main function of mitochondria, these organelles are key components of many other cellular and metabolic pathways, such as the tricarboxylic acid (TCA or Krebs) cycle, fatty acid oxidation, steroid and pyrimidine synthesis, and the urea cycle. In addition, mitochondria play pivotal roles in several processes, including apoptosis, mitophagy, and intracellular calcium homeostasis.

Mitochondrial diseases are the most frequent inborn errors affecting metabolism, with an estimated prevalence of between 5 and 15 cases per 100,000 individuals [15]. Although extremely heterogeneous from the clinical, biochemical, and genetic points of view, these disorders are all characterized by a dysfunctional OXPHOS system [16], which leads, in most cases, to neurological impairment [17]. More than 340 different genes have been described as being causative of mitochondrial disorders [16]. Mitochondria are peculiar organelles in eukaryotic cells because they contain their own genome, the mitochondrial DNA (mtDNA), which encodes for core components of the OXPHOS complexes (Figure 3). The other subunits of the OXPHOS machinery, as well as all the proteins necessary for its assembly and for the expression of the mitochondrial subunits, originate from nuclear DNA (nDNA), are synthesized in the cytosol, and are actively imported inside the organelle. Thus, mitochondrial diseases can arise from mutations in genes localized in either genome and the inheritance pattern can be either autosomal or X-linked for mutations in nDNA or maternal for mutations in mtDNA. The causes of mitochondrial disease can be classified according to the function of the product of the mutated gene [16] and include not only defects in mtDNA maintenance, mitochondrial gene expression, and synthesis of enzymatic cofactors but also in mitochondrial dynamics and quality control. However, a prominent group of genes associated with mitochondrial disease are those encoding the structural components of the OXPHOS complexes and of specific assembly factors, which are not part of the mature structures but are essential for their proper maturation [18].

## 3. Models of Mitochondrial Disease in *Drosophila melanogaster*

### 3.1. Fly Models of Complex I Defects

Complex I is the largest and most intricate of the respiratory chain enzymes. In humans, it is composed of 44 subunits, 14 of which are “core subunits”—i.e., conserved through evolution from bacteria to humans—while the rest are “supernumerary subunits”, not directly involved in catalysis but important for the stability and/or biogenesis of the enzyme [19,20]. Complex I deficiency is the most common OXPHOS defect and the majority of patients present with neurological impairment, often in the form of Leigh syndrome, with or without the involvement of other organs [21]. Mutations in all mtDNA-encoded subunits, as well as in 24 nuclear-encoded subunits, have been linked to human disease [18].

*D. melanogaster* has been used as a model to study complex I biogenesis [22,23,24,25], and, importantly, the structure of *D. melanogaster* complex I has recently been determined [26,27]. *D. melanogaster* complex I is a 43 subunit complex with high structural homology to its mammalian counterpart and basically the same subunit composition, except for NDUFC1, which is mammalian-specific, and NDUFA2, a highly conserved subunit in terms of sequence but which appears to be loosely associated with the fly complex I [26,27].

Several *D. melanogaster* models for complex I deficiency have been produced and characterized, mostly using RNAi (Table 1).

These include models for genes encoding both core [28,29,30,31] and supernumerary subunits [31,32,33,34]. Similar to findings reported in humans, flies subjected to knockdown (KD) for ND75 (NDUFS1 homolog) exhibited severe neurological impairment with reduced neuromotor function and longevity [28]. An interesting observation obtained by using cell type-specific KD is that neuronal degeneration is linked to complex I defects in glia rather than primary dysfunction in neurons [28]. Similar findings were reported using a KD model for ND23, the NDUFS8 ortholog, which also unraveled the predominant involvement of glia in the neurodegenerative process [30], despite the lack of behavioral alterations. The specific role of glia in neurological manifestations of mitochondrial disorders has not been investigated in detail in other animal models and patients, but several lines of evidence point to the importance of correct mitochondrial function in glia for neuronal physiology and survival [35].

Fly models of complex I deficiency also include a triple amino acid deletion (p.Met186_Ser188del) in the mtDNA-encoded subunit gene mt:ND2 in the homoplasmic state (mt:ND2^del1^) [29] obtained by manipulation of the mtDNA with mitochondria-targeted restriction enzymes [36]. Considering the difficulties of manipulating mammalian mtDNA, this and the other mutants for the mtDNA-encoded COX subunits (described below) are extremely relevant models of mtDNA-linked disease [37]. Notably, the mt:ND2^del1^ variant is not lethal and manifests as a hypomorphic mutation, causing mild neuromotor dysfunction and minor neurodegeneration [29]. However, this model has given important clues about complex I function, as the proton pumping activity is impaired without majorly impacting electron transfer [29].

In addition to defects in genes encoding structural subunits of complex I, other *D. melanogaster* models with defects in accessory proteins involved in complex I biogenesis (assembly factors) have been characterized, such as CIA30/NDUFAF1 and Sicily/NDUFAF6 [33,38].

These models were generated using different approaches, such as transposon mobilization, chemical mutagenesis, and the UAS/GAL4 system. Even if different genes were targeted, all of them exhibited similar phenotypic features, mainly because they commonly resulted in complex I enzymatic defects. For example, loss of function mutations and strong ubiquitous knockdowns were mostly characterized by severe phenotypes, such as developmental arrest at the larval/pupal stages. On the other hand, hypomorphic mutations and tissue-specific or mild knockdowns usually led to milder phenotypes, often resembling the clinical features observed in patients; i.e., shorter lifespan, decreased neuromotor function, neurodegeneration, seizures, myopathy, increased susceptibility to exogenous stressors, and cardiac dysfunction.

### 3.2. Fly Models of Complex II Defects

In humans, pathological variants in the four CII structural subunits (SDHA-D) and in two assembly factors (SDHAF1 and SDHAF2) have been associated with either familial tumors, such as paraganglioma and pheochromocytoma, or classical mitochondrial disease [18,39,40]. To date, fly models of three of the four structural subunits of succinate dehydrogenase (SDHA, SDHB, and SDHC) have been produced [41,42,43,44,45]. In addition, mutants in two assembly factors named the Sdhaf3 and Sirup/Sdhaf4 homologs, respectively [45,46], have been characterized (Table 1). Complex II deficiency in flies causes typical mitochondrial dysfunction-associated neurological phenotypes, such as diminished climbing ability, abnormal wing posture, and neurodegeneration, as well as reduced lifespan. Notably, a feature that is frequently observed in complex II deficiencies in flies is hypersensitivity to O_2_ and increased susceptibility to oxidative stress, with subsequent oxidative damage to proteins [41,43,44,45,46]. An important difference between humans and flies is that, while SDH variants have often been linked to different forms of malignant paragangliomas in humans [47], no evidence has been reported in *Drosophila* models of CII defects.

### 3.3. Fly Models of Complex III Defects

Among the OXPHOS defects in human patients, complex III deficiency is the rarest [48]. Using different genetic approaches (i.e., gene KD and KO), three models of complex III deficiency targeting the fly homologues of TTC19, BCS1L [49,50,51], and UQCR10 [52] have been generated and characterized (Table 1). *BCS1L* and *TTC19* pathological variants constitute the most frequently found genetic defects in mitochondrial disease associated with isolated complex III deficiency [48]. Similarly to humans, *Ttc19* defects in flies cause a chronic, non-lethal form of neurological CIII deficiency [49,50]. In contrast, *Bcs1* knockdown has rather severe effects on *D. melanogaster* development, as individuals arrest at the larval stage without growing, most probably due to severe CIII deficiency [51]. The fact the partial loss (KD) of the gene causes such a strong phenotype in *D. melanogaster* is compatible with the fact that, so far, only missense and no loss-of-function mutations have been reported in *BCS1L*-linked human disease [48].

Notably, the brain-specific silencing of *Bcs1* in *D. melanogaster* allows the larvae to grow and pupate. However, most of the flies die at the pupal stage and, if some individuals survive to adulthood, they suffer from severe paralysis and die in a few days. In contrast, the specific silencing in skeletal muscle leads to complete lethality at the pupal stage [51]. It is important to note that growth retardation, aminoaciduria, cholestasis, iron overload, lactic acidosis, and early death (GRACILE) syndrome is a very severe autosomal recessive human condition linked to one specific BCS1L mutation (p.Ser78Gly) [53]. In contrast to other *BCS1L* pathological variants, liver failure seems to be a determinant component for the early-onset lethality of GRACILE syndrome [48,54]. However, knockdown of *Bcs1* in the fat body, the insect tissue that functionally resembles both the mammalian liver and adipose tissue, has milder effects on *D. melanogaster* fitness, causing only a slight reduction in lifespan without impacting development [51]. Thus, understanding physiological differences between humans and flies and species-specific features might explain why liver disease is very severe in some specific forms of syndromes caused by BCS1L deficiency.

The third *D. melanogaster* model of complex III deficiency is linked to a defect in *oxen* (*UQCR10* homolog), a gene that is most likely related to severe cases of in utero onset of ventriculomegaly, apnea, developmental regression, hypotonia, and seizure [55]. Similarly, *ox* mutants are affected by lethality at the first larval stages [52]. Finally, two neuronal peptides (named *sloth1 and sloth2* in *D. melanogaster*) originating from a bicistronic transcript were linked to complex III biogenesis in flies [56]. Interestingly, sloth1 and sloth2 are homologs of two recently identified mammalian complex III assembly factors named SMIM4 and Brawnin, respectively [57,58]. Complete loss and ubiquitous RNAi of *sloth1* and *sloth2* cause developmental lethality and neurodegeneration in escaping adults [56].

### 3.4. Fly Models of Complex IV Defects

Deficiencies in the terminal oxidase of the mitochondrial respiratory chain—i.e., cytochrome c oxidase (COX) or complex IV—are a major cause of mitochondrial disease in humans [59]. Isolated COX deficiencies are mostly associated with mutations in a large number of genes encoding COX structural subunits (either mtDNA- or nDNA-encoded) or, most frequently, assembly factors. COX deficiency is also a feature in patients with mutations in genes encoding mitochondrial gene expression factors, such as LRPPRC, a mitochondrial RNA stabilizing factor; TACO1, a specific translational activator of MT-CO1; or even mitochondrial tRNAs and aminoacyl-tRNA-synthetases [59]. Mutations in nucleus-encoded structural subunits were hypothesized to be embryonic-lethal for a long time because none were found until 2008, when mutations in *COX6B1* were identified [60]. After that, several other mutations in other nDNA-encoded genes encoding different complex IV structural subunits were described, but the quantity of disease-related genes encoding COX assembly factors outnumbers the former by far. In the case of COX deficiency, the spectrum of clinical presentations is extremely heterogeneous and ranges from encephalopathic syndromes to cardiomyopathies [59,61,62]. The most frequent presentation of COX deficiency is Leigh syndrome, associated with mutations in *SURF1*, which encodes an assembly factor with a still-unclear function [63]. COX is highly conserved between humans and flies, with all the 14 subunits composing mammalian COX complex being present in flies, including a COX7B ortholog, which was initially thought to be missing in insects [64]. Missense mutations in two of the three mtDNA-encoded COX subunits (i.e., *mt:CoI* and *mt:CoII*, the genes encoding the two catalytic subunits) have been described in the homoplasmic state in *D. melanogaster* [36,65]. Depending on the mutation, flies displayed a wide range of phenotypes, from healthy (silent) mutations (as in the case of the p.Ala302Thr mutation in *mt:CoI*) to harmful mutations specific to males (leading to male sterility, such as the p.Arg301Leu mutation) and more severe mutations (such as p.Arg301Ser) triggering growth retardation and neurodegeneration. In addition, numerous models of COX deficiency linked to either defects in nuclear DNA-encoded subunits or assembly factors have been generated and characterized (Table 1). In the early 2000s, mutations in the supernumerary subunits *COX5A*, *levy/COX6A*, and *cype/COX6C* [66,67,68] were introduced.

A few years later, RNAi models for genes encoding the subunits *COX4*, *COX5A*, *COX5B*, *levy/COX6A*, *COX6B*, *cype/COX6C*, and *COX7A* were described [68,69,70], as well as, more recently, *COX7B* [64]. Most of these models cause very severe pleiotropic phenotypes, often resulting in developmental lethality and systematically causing neurodegeneration when the RNAi is restricted to the central nervous system.

Several homologs of COX assembly factors have been studied in flies (Table 1). These include *Ccdc56/Coa3* [71], the single *SCO1* and *SCO2* homolog *Scox* [72,73,74], *Surf1* [75,76], and the more recently identified genes encoding metazoan-specific assembly factors *Coa7* [77] and *Coa8* [78].

Compound heterozygous mutations in *COA3* have been identified in one human subject presenting neuropathy linked to COX deficiency [79]. Ccdc56/Coa3 is essential in flies because its complete loss hampers development, causing growth arrest at the larval stage [71]. Ubiquitous RNAi with *Surf1* in *D. melanogaster* is also linked to a severe phenotype and developmental arrest [75,76]. The developmental phenotype of *Surf1* RNAi flies is also severe when restricted to muscle [75,76]. Even if *SURF1* loss-of-function mutations in humans are associated with severe early-onset encephalopathy, neuronal-specific silencing of *Surf1* led to a milder phenotype, with normal development and no major signs of neuropathology. However, slightly decreased neuromotor function was still observed in these flies [75].

A similar observation was recently reported for *Scox* defects, as neuron-specific knockdown seemed to have little effect on *D. melanogaster* development and behavior whereas glial KD caused severe deterioration of the neuromotor function [80].

Thus, COX deficiency in flies mimics the human phenotype well, ranging from severe manifestation and early death to neurological disorder. Importantly, cholinergic and adrenergic neurons have been demonstrated to be highly sensitive to COX deficiency in flies, whereas dopaminergic neurons are not.

Recently, by using a set of fly models with KD expression of structural COX subunits (*cype*) and assembly factors (*Coa8*, *Coa3*, and *Scox*), it was demonstrated that COX defects lead to altered cellular homeostasis and compartmentalization of transition metals; in particular, copper [81]. The contributions of these alterations to the pathogenesis of human diseases warrant more investigation.

### 3.5. Fly Models of Complex V Defects

The majority of patients with complex V deficiency harbor mutations in the mtDNA region encoding the MT-ATP6 subunit, causing two main phenotypes: either Leigh syndrome or neuropathy, ataxia, and retinitis pigmentosa (NARP) syndrome. However, mutations in the other CV mtDNA-encoded subunit, MT-ATP8, have also been described. Only a few cases of nuclear genes have been identified in patients with complex V deficiency, principally associated with encephalopathic syndromes [18]. These can either encode structural subunits, such as *ATP5F1A*, *ATP5F1D*, and *ATP5F1E*, or assembly factors; namely, *ATPAF2* and *TMEM70*, the latter now considered as an assembly factor for both complex V and complex I [82,83].

A point mutation (p.Gly116Glu) in the mitochondrially encoded *mt:ATPase6* gene (*MT-ATP6* homolog) resulting in complex V deficiency was found in flies [84]. This was a spontaneous mutation that was identified in the homoplasmic state in flies suffering from a maternally inherited neurodegenerative phenotype and shorter lifespan.

More recently, different genetic manipulation approaches have been exploited to study the effects in flies of defects in *ATPsynB*, encoding subunit b (*ATP5PB* in humans); *ATPsynC* (*ATP5MC1* homolog); and *ATPsynD*, encoding subunit d (the *ATP5PD* homolog) [85,86,87] (Table 1). Different *ATPsynC* alleles of varying severity were generated via transposon mobilization and chemical mutagenesis [86]. Null alleles were developmentally lethal whereas hypomorphic alleles caused phenotypes ranging from growth retardation and severe lifespan reduction to hypoactivity and neuromotor dysfunction [86].

Ubiquitous knockdown of *ATPsynB* and *ATPsynD* genes resulted in growth arrest and developmental lethality before pupation. Notably, sole misexpression of *ATPsynB* in testes allowed development but severely impaired fertility in males [87]. In this regard, it is important to note that work undertaken using a set of RNAi targeting *D. melanogaster* MRC subunits demonstrated that ATP synthase defects in the germline impact differentiation through a mechanism that is independent from OXPHOS dysfunction [88]. The mechanistic details, however, warrant future work.

### 3.6. Coenzyme Q Deficiency Models

Primary coenzyme Q (CoQ) deficiencies constitute a group of mitochondrial diseases caused by mutations in genes encoding some of the enzymes involved in the synthesis pathway of this essential lipid [89]. As with other mitochondrial diseases, coenzyme Q deficiencies are genetically and clinically extremely heterogeneous. However, the involvement of the CNS in this group of disorders is also very prominent. Specifically, encephalopathy and Leigh-like signs are often present in CoQ deficiencies and typically associated with developmental delay, neuromotor dysfunction, and epilepsy [89]. Defective biosynthesis of CoQ in *D. melanogaster* has been investigated by studying mutations in *qless*, the *PDSS1* ortholog [90] (Table 1). Despite the fact that some forms of CoQ deficiencies also manifest with a renal phenotype, *qless* mutation in *D. melanogaster* leads to severe specific defects in the CNS, with increased caspase activation and neuronal death, similarly to most of the human cases reported with mutations in CoQ-related genes [89].

**Table 1 biomolecules-13-00378-t001:** Fly models of respiratory chain defect.

	Fly Gene	Human Ortholog	Function	System	Tissue Specificity	Phenotype	Ref.
**Complex I**	mt:ND2	MT-ND2	Core subunit	Restriction enzymes targeting mtDNA	Ubiquitous	Neuromotor dysfunction, neurodegeneration	[29]
	ND-75	NDUFS1	Core subunit	RNAi	Glia	Neurodegeneration	[28]
				RNAi	Ubiquitous	Neurodegeneration	
				RNAi	Neurons	Reduced lifespan	
	ND-23	NDUFS8	Core subunit	RNAi	Glia	Neurodegeneration	[30]
				RNAi	Ubiquitous	Developmental arrest	
				RNAi	Neurons	Reduced lifespan, neuromotor dysfunction	
	ND-20	NDUFS7	Core subunit	RNAi	Ubiquitous	Array of phenotypes depending on RNAi efficiency	[31]
	ND-51	NDUFV1	Core subunit	RNAi	Ubiquitous	Developmental arrest	
	ND-19	NDUFA8	Supernumerary subunit	RNAi	Ubiquitous	Developmental arrest	[32]
	ND-39	NDUFA9	Supernumerary subunit	RNAi	Ubiquitous	Developmental arrest	[32]
	ND-42	NDUFA10	Supernumerary subunit	RNAi	Ubiquitous	Developmental arrest	[33]
				RNAi	Eye	Retinal degeneration	
	Sicily	NDUFAF6	Assembly factor	FLP/FRT system	Mosaic eye	Retinal degeneration, neurodegeneration	[33]
				Transposable elements	Ubiquitous	Developmental arrest	
	ND-18	NDUFS4	Supernumerary subunit	RNAi	Ubiquitous	Array of phenotypes depending on RNAi efficiency	[31]; [34]
	CIA30	NDUFAF1	Assembly factor	Transposable elements	Ubiquitous	Developmental arrest	[38]
				RNAi	Ubiquitous	Reduced growth, partial developmental lethality	
**Complex II**	SdhA	SDHA	Subunit	FLP/FRT system	Mosaic eye	Retinal degeneration	[41]
				FLP/FRT system	Ubiquitous	Developmental arrest	
	SdhB	SDHB	Subunit	Transposable elements	Ubiquitous	Reduced lifespan, sensitivity to hyperoxia, age-related neuromotor dysfunction	[43]
	SdhC	SDHC	Subunit	Overexpression of dominant negative mutation	Neuronal	Reduced lifespan, oxidative damage	[44]
	Sirup/Sdhaf4	SDHAF4	Assembly factor	TALENs	Ubiquitous	Reduced lifespan, neurodegeneration, sensitivity to oxidative stress	[45]
	Sdhaf3	SDHAF3	Assembly factor	Homologous recombination	Ubiquitous	Sensitivity to oxidative stress and hyperoxia, age-related neuromotor dysfunction	[46]
	Ttc19	TTC19	Assembly factor	Transposable elements	Ubiquitous	Neuromotor dysfunction	[49]
				CRISPR/Cas9 KO	Ubiquitous	Neuromotor dysfunction	[50]
	Bcs1	BCS1L	Assembly factor	RNAi	Ubiquitous	Developmental arrest, larval neuromotor dysfunction	[51]
				RNAi	Neurons	Reduced lifespan, neuromotor dysfunction, paralysis	
				RNAi	Muscle	Developmental arrest	
				RNAi	Fat body	Reduced lifespan	
	Ox	UQCR10	Supernumerary subunit	Transposable elements	Ubiquitous	Developmental arrest	[52]
	sloth1	SMIM4/ UQCC5	Assembly factor	RNAi	Ubiquitous	Developmental lethality, neurodegeneration	[56]
				CRISPR/Cas9 KO	Ubiquitous (somatic)	Developmental lethality, neurodegeneration	
				CRISPR/Cas9 KO	Ubiquitous (germline)	Developmental lethality, neurodegeneration	
	sloth2	Brawnin/ UQCC6	Assembly factor	RNAi	Ubiquitous	Developmental lethality, neurodegeneration	[56]
				CRISPR/Cas9 KO	Ubiquitous (somatic)	Developmental lethality, neurodegeneration	
				CRISPR/Cas9 KO	Ubiquitous (germline)	Developmental lethality, neurodegeneration	
**Complex IV**	mt:CoI	MT-CO1	Core subunit	Mitochondrially targeted restriction enzymes	Ubiquitous	Reduced growth, neurodegeneration	[36]
	COX7B	COX7B	Supernumerary subunit	RNAi	Ubiquitous	Developmental arrest	[64]
	cype/ COX6C	COX6C	Supernumerary subunit	FLP/FRT system	Eye	Retinal degeneration	[66]
				FLP/FRT system	Germline	Developmental arrest	
				RNAi	Ubiquitous	Developmental arrest	[81]
	COX5A	COX5A	Supernumerary subunit	FLP/FRT system	Eye	Retinal degeneration	[67]
				RNAi	Ubiquitous	Developmental arrest	[70]
	levy/ COX6A	COX6A1	Supernumerary subunit	Chemical mutagenesis	Ubiquitous	Temperature-induced paralysis, bang-induced paralysis, neurodegeneration, reduced lifespan	[68]
				RNAi	Ubiquitous	Developmental lethality	[70]
	COX4	COX4I1	Supernumerary subunit	RNAi	Ubiquitous	Developmental arrest (strong RNAi), reduced lifespan (mild RNAi)	[69]
	COX5B	COX5B	Supernumerary subunit	RNAi	Ubiquitous	Developmental arrest	[69]
				RNAi	Ubiquitous	Developmental arrest	[70]
	COX7A	COX7A1	Supernumerary subunit	RNAi	Ubiquitous	Developmental arrest	[70]
	Ccdc56/Coa3	COA3	Assembly factor	Transposable elements	Ubiquitous	Developmental arrest	[71]
	Scox	SCO1/ SCO2	Assembly factor	Transposable elements	Ubiquitous	Developmental arrest	[72]
				RNAi	Ubiquitous	Developmental arrest	[73]
				RNAi	Heart	Reduced lifespan, cardiac dysfunction	[74]
				RNAi	Glia	Neuromotor dysfunction	[80]
	Surf1	SURF1	Assembly factor	RNAi	Ubiquitous	Developmental arrest	[75]; [76]
				RNAi	Neurons	Mild neuromotor defects	[75]
				RNAi	Muscle	Developmental arrest	[76]
	Coa7	COA7	Assembly factor	RNAi	Eye	Retinal degeneration	[77]
				RNAi	Neurons	Reduced lifespan, neuromotor dysfunction	
	Coa8	COA8	Assembly factor	RNAi	Ubiquitous	Sensitivity to oxidative stress, neuromotor dysfunction	[78]
				RNAi	Neurons	Sensitivity to oxidative stress, neuromotor dysfunction	
**Complex V**	mt:ATPase6	MT-ATP6	Core subunit	Isolation of spontaneous mutation	Ubiquitous	Reduced lifespan, progressive neurodegeneration	[84]
	ATPsynD	ATP5PD	Core subunit	RNAi	Ubiquitous	Developmental arrest	[85]
	ATPsynB	ATP5PB	Core subunit	RNAi	Ubiquitous	Developmental arrest	[87]
	ATPsynC	ATP5MC1/ATP5MC2/ATP5MC3	Core subunit	Transposable elements, chemical mutagenesis	Ubiquitous	Range of phenotypes depending on the severity of the genetic lesion	[86]
**CoQ**	qless	PDSS1	CoQ biosynthesis	Chemical mutagenesis	Ubiquitous	Developmental arrest	[90]
				FLP/FRT system	Neurons	Neurodegeneration	

### 3.7. Defects in Mitochondrial DNA Replication and Maintenance

So far, numerous genes have been linked to mtDNA replication and maintenance defects in human disease [91]. These include genes encoding factors that are directly dedicated to replication of the mitochondrial genome, such as *POLG*, *POLG2*, *TWNK*, and *TFAM*, and those indirectly involved in the maintenance of mtDNA, such as enzymes involved in dNTP synthesis (e.g., *TK2*, *DGUOK*, *SUCLG1*, and *SUCLA2).* Other genes associated with mtDNA instability have unknown functions (e.g., *MPV17*). It is important to note that mutations in genes encoding proteins involved in mitochondrial dynamics (e.g., *OPA1* and *MFN2*) can also cause mtDNA maintenance disorders, as proper mitochondrial architecture seems to be essential for correct mtDNA replication [92].

Defects in *D. melanogaster* POLγ, the mtDNA-specific DNA polymerase, were first reported in 1999 [93]. In fact, the gene encoding the catalytic subunit (subunit α) of mtDNA polymerase, initially named *tamas* (the Sanskrit word for “darkness”)—official symbol *PolG1*—was identified during a screening of pupal lethal phenotypes. Numerous pathogenic alleles of *PolG1* have been described since then, most of them affecting viability at or before the pupal stage [93,94,95] (Table 2).

Importantly, *D. melanogaster* POLγ has been engineered to generate models making it possible to study the effects of random generation and accumulation of mtDNA mutations in vivo (mtDNA mutator models). Firstly, the exonuclease domain of *PolG1* was mutated to impair the proofreading activity of the enzyme, and this mutant was used to complement a *PolG1* KO strain [94]. Homozygosity in proofreading defective *PolG1* (named the *exo*^−^ allele) causes developmental lethality in *D. melanogaster*, but heterozygous individuals do not show behavioral defects, despite having increased mutational rates in mtDNA throughout the generations [94]. In addition, a second mutator fly model carrying the very same mutation in the proofreading domain of POLγ was generated using a different approach, which was transgenic expression of exo^−^ PolG1 [96]. Notably, in this work, the authors noted some differences between the two mutator fly models. In fact, while the first model was lethal in homozygosity [94], this was not observed in the second model [96]. The causes behind these discrepancies are currently unclear, but they might be explained by differing mutational heterogeneity between the two models. Further, and importantly, mtDNA heteroplasmy levels are likely to have a primary modifying role. In fact, studies using the analogous mutator mouse model have also led to an intense debate regarding the role of mtDNA mutations in disease and aging [97,98,99]. It is worth mentioning that an alternative approach for generating a *D. melanogaster* mtDNA mutator model was based on mitochondrial targeting of APOBEC1, a vertebrate cytidine deaminase enzyme [100]. This enzyme can specifically introduce point mutations that do not affect the mtDNA copy number, introduce insertions/deletions, or affect development. However, the accumulation of mtDNA mutations did cause early death and mitochondrial dysfunction in the adult stage.

Fly disease models of the mtDNA-helicase gene (mammalian *TWNK*) have also been generated and studied (Table 2). Firstly, three mtDNA-helicase variants corresponding to human autosomal dominant PEO mutations were expressed in vivo [101]. Two of them (p.Lys388Ala and p.Ala442Pro) caused mtDNA depletion and severe phenotypes, resulting in arrest at different developmental phases before the adult stage. Curiously, the third dominant mutation (p.Trp441Cys) did not show strong effects, as mtDNA depletion levels were minimal and no developmental arrest was observed. In addition, RNAi was used to perturb mtDNA-helicase gene expression [95]. Similar to the effect of the overexpression of dominant negative mutants, KD of the helicase-encoding gene resulted in mtDNA depletion and lethality of around 75% in individuals at the pupal stage.

Recently, variants of *bor* (*belphegor*), the homolog of *ATAD3A*, a gene associated with mitochondrial disease in humans and encoding a component of the nucleoid (i.e., the association of mtDNA and proteins) [102,103,104], have been studied in *D. melanogaster* [103,105]. In these cases, the phenotypes observed in flies harboring missense pathogenic variant were compatible with mitochondrial disease and included hypotonia, developmental delay, cardiomyopathy, and brain abnormalities [105], whereas complete loss of *bor* had been previously linked to growth arrest at the larval stage [106].

Mutations in *SUCLG1*, encoding the alpha subunit of the succinyl-CoA synthetase, cause severe early-onset mtDNA depletion syndromes in humans [107,108,109]. In contrast, loss of function in the fly homolog Scsα1 does not lead to early lethal phenotypes. However, disease phenotypes, such as developmental delay, altered locomotor behavior, and reduced lifespan under starvation, were still observed [110].

Very recently, a neuron-specific RNAi *Drosophila* model of *MPV17* (*dMpv17*) was reported and showed impaired locomotor activity in larvae and learning ability in adults, altered energy metabolism, and abnormal neuromuscular junctions [111]. This is an interesting observation, as patients, who were characterized by early-onset liver failure due to profound depletion of mtDNA in the liver, developed a progressive neurological phenotype at later stages [112]. In addition, peripheral neuropathy has been reported for some patients [113].

### 3.8. Defects in Mitochondrial Gene Expression

Mitochondria contain separated gene expression machineries for the synthesis of the mtDNA-encoded polypeptides; i.e., specific mitochondrial transcription and translation factors (nDNA encoded), mtDNA-encoded transfer RNAs (tRNAs), and mitochondrial ribosomes (mitoribosomes) composed of nDNA-encoded proteins and of ribosomal RNAs (rRNAs) encoded in the mtDNA [114]. In the last few years, many factors involved in mitochondrial gene expression—in particular, translation—have been linked to human disorders, including mutations in mitoribosomal proteins [115,116]. Most of the disorders linked to mitochondrial gene expression have neurological manifestations, such as leukoencephalopathy and Leigh syndrome [115,116].

Notably, the first *D. melanogaster* model of mitochondrial dysfunction was a mitochondrial ribosomal protein mutant. This model was reported in 1987 when a pathological variant of the technical knockout (*tko*) gene was found in homozygosity in flies suffering from a neurological temporary paralytic phenotype induced by mechanical shock known as “bang sensitivity” [117]. The gene was found to encode the mitochondrial ribosomal protein S12 (mRpS12). Later, the *tko* fly model was further studied as a model of mitochondrial disease. Indeed, in addition to bang sensitivity, mutant flies were found to suffer from developmental delay, hypersensitivity to doxycycline (an inhibitor of mitochondrial translation), and deafness due to mitochondrial respiratory chain dysfunction [118]. More recently, neuronal-specific RNAi of *D. melanogaster* genes encoding the mitochondrial ribosomal proteins mRpL15 and mRpL40 showed disruption of synapse development and function [119]. Thus, altered mitochondrial translation also predominantly causes neurological phenotypes in *D. melanogaster* (Table 2).

### 3.9. Defects in Mitochondrial Dynamics and Architecture

In recent decades, interest in the influence of mitochondrial architecture and dynamics on health and disease has increased considerably. Mitochondria are not static and isolated organelles but instead form a highly dynamic “mitochondrial network” governed by fission and fusion processes [120]. In fact, proper “mitodynamics” appears to be very relevant for different processes related to mitochondrial function, such as mtDNA replication, metabolism, and recycling of dysfunctional/damaged mitochondria [120]. Moreover, proper mitodynamics is important for organism development [121,122,123,124]. As a consequence, human mitochondrial disorders can also be caused by dysfunctional components controlling fission and fusion, such as OPA1, MFN1-2, MFF, and DRP1 [125].

Along with studies performed in vitro, several animal models, including *D. melanogaster* models (Table 2), have been generated to study the effects of defective mitochondrial morphology and dynamics in vivo. Several mutations in fly *Drp1* lead to developmental lethality at the third larval stage, associated with impaired neurotransmission [126], and the specific loss of *Drp1* in spermatocytes leads to altered spermatogenesis due to mitochondrial clustering and impaired motility [127]. Essential factors for maintaining mitochondrial network morphology and cristae shape include OPA1 and MFN2, the dysfunction of which causes an array of phenotypes in *D. melanogaster* linked to defective mitochondrial architecture, including developmental delay or arrest, cardiomyopathy, and neurological phenotypes resembling autosomal dominant optic atrophy (ADOA) and Charcot–Marie–Tooth type 2A syndrome (CMT2A) [128,129,130,131].

In addition, the inner mitochondrial membrane ultrastructure is intimately related to mitodynamics because it depends on the functions of proteins such as OPA1 or DRP1. However, mitochondrial ultrastructure is also heavily influenced by other factors, such as the dimerization of complex V (ATP synthase) at the *cristae* rims [132,133] and the mitochondrial contact sites and cristae organization system (MICOS) complex [134].

Mutations in some MICOS components have been reported in humans, including MIC13/QIL1 [135,136,137], causing a severe form of infantile hepato-encephalopathy; MIC26/APOO [138], associated with an X-linked mitochondrial myopathy with cognitive impairment; and MIC60, linked to Parkinsonism [139]. In *D. melanogaster*, deletion of the main component of the MICOS complex—MIC60/Mitofilin—causes a severe developmental phenotype with growth arrest at the pupal stage [139,140]. However, *APOO* loss in *D. melanogaster* is associated with milder phenotypes, partial developmental lethality, and mitochondrial ultrastructure defects with multiple OXPHOS deficiencies [138]. Notably, MIC13/QIL1 depletion in muscle and neurons causes abnormal mitochondrial network, ultrastructure, and function [141].

Finally, a member of the solute carrier family (named *SLC25A46*) has repeatedly been reported to be associated to different forms of neurological mitochondrial disorder and Leigh syndrome [142,143,144,145]. *SLC25A46* encodes a mitochondrial outer membrane protein involved in mitochondrial dynamics that interacts with MFN2, OPA1, and MICOS [143]. A *D. melanogaster* model for Slc25A46a was recently described [146]. Specifically, Slc25A46a knockdown in fly neurons causes neurological phenotypes both in larvae and adults, with reduced neuromotor function and altered morphology in the neuromuscular junction [146].

**Table 2 biomolecules-13-00378-t002:** Fly models of other mitochondrial defects.

	Fly Gene	Human Ortholog	Function	System	Tissue Specificity	Phenotype	Ref.
**mtDNA replication and maintenance**	PolG1/tam	POLG	mtDNA replication	Chemical mutagenesis	Ubiquitous	Developmental arrest, neuromotor dysfunction	[93]
				Homologous recombination	Ubiquitous	Developmental arrest, reduced growth	[94]
				RNAi	Ubiquitous	Developmental arrest	[95]
				KI of PolG1 exo^−^ (mutator)	Ubiquitous	Developmental lethality in homozygosity, increased mtDNA mutation rate in heterozygosity	[94]
				Transgenic PolG1 exo^−^ (mutator)	Ubiquitous	Reduced lifespan, dose-dependent increase in mtDNA mutation rate	[96]
				RNAi	Ubiquitous	Partial developmental lethality	[95]
	mtDNA-helicase	TWNK	mtDNA replication	Transgenic expression of dominant mutations	Ubiquitous	Developmental arrest and mtDNA depletion	[101]
	bor	ATAD3A	Component of nucleoids	Transgenic expression of dominant mutation	Ubiquitous	Developmental arrest	[103]
					Neurons	Developmental arrest	
					Muscle-specific	Partial developmental lethality	
				Transposable elements	Ubiquitous	Developmental arrest	[106]
	SCSα1	SUCLG1	Mitochondrial nucleotide synthesis	CRISPR/Cas9	Ubiquitous	Developmental delay, altered neuromotor function, and reduced lifespan under starvation	[110]
**Mitochondrial translation**	tko	MRPS12	Mitoribosome small subunit	Chemical mutagenesis	Ubiquitous	Bang-induced paralysis, developmental delay, sensitivity to doxycycline	[117]; [118]
	mRpL15	MRPL15	Mitoribosome large subunit	RNAi	Neurons	Disruption of synapse development and function	[119]
	mRpL40	MRPL40	Mitoribosome large subunit	RNAi	Neurons	Disruption of synapse development and function	[119]
**Mitochondrial dynamics/architecture**	Drp1	DRP1	Mitochondrial fission	Transposable elements	Ubiquitous	Partial developmental lethality, altered neuromotor function	[126]
				FLP/FRT system	Spermatocytes	Altered spermatogenesis and sperm motility	[127]
	Opa1	OPA1	Mitochondrial fusion	FLP/FRT system	Eye	Retinal degeneration	[128]
				Transposable elements	Ubiquitous	Developmental arrest	[128]
				RNAi	Heart	Cardiomyopathy	[131]
	Marf	MFN1/2	Mitochondrial fusion	RNAi	Heart	Cardiomyopathy	[131]
	Mic26-27	MIC26-27	Cristae architecture	Transposable elements	Ubiquitous	Partial developmental lethality, reduced lifespan, reduced neuromotor function	[138]
	Mitofilin	IMMT/ MIC60	Cristae architecture	Transposable elements	Ubiquitous	Developmental arrest	[139]
				RNAi	Muscle	Mild neuromotor defects	
				RNAi	Neurons	Mild neuromotor defects	
	Slc25A46a	SLC25A46	Mitochondrial dynamics	RNAi	Neurons	Neuromotor dysfunction	[146]

## 4. Conclusions

The growing number of *Drosophila melanogaster* models of mitochondrial deficiency underscores their usefulness in the study of the phenotypical, biochemical, and molecular features of human mitochondrial diseases. As we described in this review, in many cases, specific genetic defects leading to OXPHOS deficiency result in observable pathological phenotypes resembling the main clinical features of patients. Notably, while the mouse models of mitochondrial disease often poorly reproduce the neurological signs typical of the human disease, flies usually show neurological phenotypes, and the study of several *Drosophila* models of mitochondrial dysfunction unraveled the central role of glia in the development of neurological phenotypes. This will open the ground for future investigations to address the pathological role of the glia in mammalian models and mitochondrial disease patients. Therefore, generating and studying these fruit fly strains has provided a key instrument not only for the validation of the pathological significance of the genetic variants found in human patients but also for the understanding of the basic cellular and molecular mechanisms related to mitochondrial diseases. The main advantages of using *D. melanogaster* models for these investigations are the easy genetic manipulation and short generation times. In fact, genetic knockdown by RNAi, which is easily and routinely applied in flies, provides a system that better resembles the situation of hypomorphic alleles, which is more frequently encountered in human mitochondrial disorders than total KO or loss-of-function mutations. A limitation of *D. melanogaster* is that, in several cases, mutations associated in humans with post-natal diseases cause developmental arrest in flies, probably due to the high energy requirements and peculiar metabolism during larva-to-pupa and pupa-to-adult transitions.

Furthermore, in practical terms, having established reliable models, *D. melanogaster* can be used as a valuable and cost-effective—but still complex—animal model that can complement the observations obtained using other models, such as murine models, which are subject to tighter ethical regulations and are much more costly timewise and economically. In addition, mice models frequently enough do not closely recapitulate human diseases. Therefore, due to these reasons, the generation of fly models can facilitate several types of translational studies, such as medium-scale drug screenings, which are necessary in order to find efficient therapies for mitochondrial diseases [147].

In conclusion, ease of handling and the low requirements for equipment and funding to carry out studies make *D. melanogaster* an attractive system for biomedical research and, more specifically, for investigations into genetic disorders, such as mitochondrial diseases.

## Figures and Tables

**Figure 1 biomolecules-13-00378-f001:**
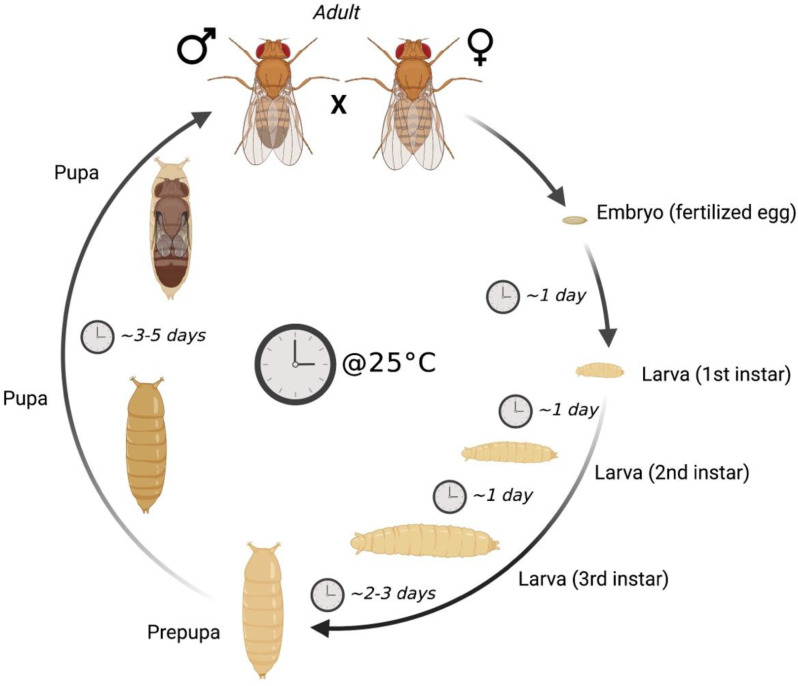
***D. melanogaster* life cycle at 25 °C.** After mating between adult female and male flies, fertilized eggs are laid and the embryo develops into a first instar larva in ~24 h. Afterwards, the larvae grow and go through two additional larval stages (second and third instars), each one lasting ~24 h. During the larval stages, *D. melanogaster* exhibits high glycolytic flux, lactate production, and a high rate of glycogen synthesis and triglyceride (TAG) accumulation, which are needed for the metamorphosis. At the end of the third instar (2–3 days), larvae pupate. During the pupal stage, metamorphosis occurs (3–5 days) and adult fly tissues form. At the end of the metamorphosis, eclosion from the puparium occurs and adult flies become fertile after ~24 h. Flies live for 60–90 days depending on the rearing conditions (i.e., temperature and diet composition).

**Figure 2 biomolecules-13-00378-f002:**
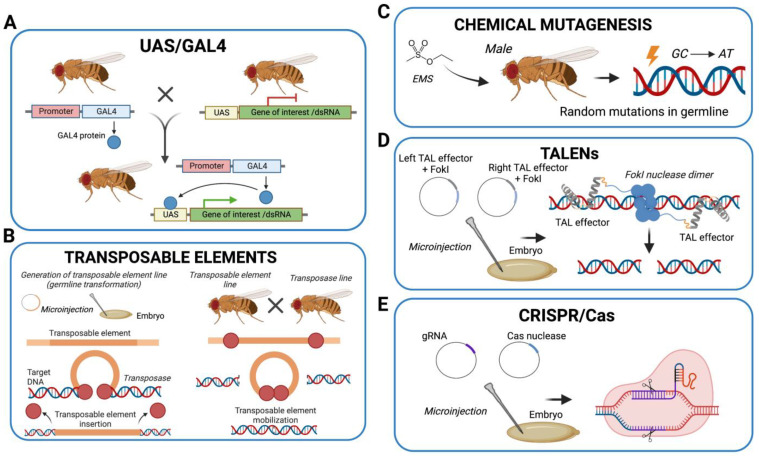
**Schematic representation of the most widely applied tools for the generation of *D. melanogaster* models of human disease.** (**A**) The GAL4/UAS system is based on crossing between a driver (GAL4) line, expressing the GAL4 transcriptional activator under the control of an endogenous *D. melanogaster* promoter, and a responder (UAS) line, expressing a construct of interest (transgene or inverted-repeat sequence for RNAi). The progeny carries both constructs and the GAL4 activator protein binds the UAS to drive the expression of the downstream construct of interest. (**B**) Flies carrying transposable elements generated through germline transformation (microinjection of an embryo) can be crossed with the transposase flies expressing the transposase enzyme that will mobilize the transposable elements in the transformed flies. Excision can be precise (rescue of the endogenous locus) or imprecise (generation of new alleles). (**C**) Chemical mutagenesis in *D. melanogaster* is achieved by treating males with mutagens (e.g., EMS), which introduce GC to AT transitions in the germline. (**D**) TALEN genome editing is based on co-injection into the embryo of two vectors carrying two TALEN constructs (left/right TAL effectors in fusion with FokI nuclease). (**E**) CRISPR/Cas editing is based on co-injection of one vector carrying gRNA constructs and one vector carrying a Cas nuclease. Abbreviations: UAS—upstream activating sequence, dsRNA—double-strand RNA, EMS—ethyl methanesulfonate, TALEN—transcription activator-like effector nuclease, CRISPR—clustered regularly interspaced short palindromic repeats, gRNA—guide RNA, Cas—CRISPR-associated protein.

**Figure 3 biomolecules-13-00378-f003:**
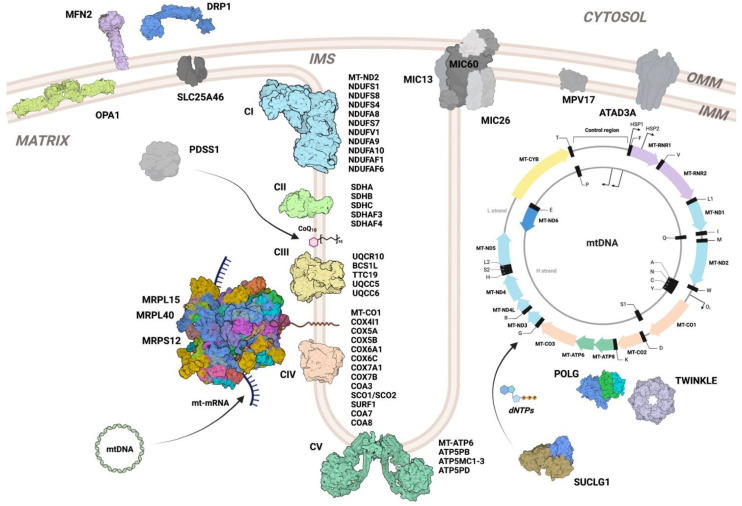
**Mitochondrial proteins and complexes for which *D. melanogaster* models are currently available.***D. melanogaster* models of mitochondrial diseases include defects in OXPHOS subunits and assembly factors (listed in Table 1), defects in mtDNA maintenance (i.e., mtDNA replication machinery, nucleoid structure, and dNTP synthesis (POLG, TWINKLE, SUCLG1, MPV17, and ATAD3A)), disorders affecting mitochondrial gene expression (mitoribosomal proteins MRPS12, MRPL15, and MRPL40), and disorders affecting mitochondrial dynamics and architecture (OPA1, DRP1, MFN2, SLC25A46, and MICOS components MIC60, MIC26/27, and MIC13). Proteins are depicted using mammalian protein structures retrieved from the following PDB IDs: complex I (5LC5), complex II (1ZOY), complex III (1BGY), complex IV (2OCC), complex V (7AJD), mitoribosome (7A5F), POLG (3IKM), Twinkle (7T8C), SUCLG1 (1EUC), s-OPA1 (6QL4), MFN2 (6JFK), and DRP1 (3ZVR). No structural data are available for the proteins depicted by generic shapes in grey.

## Data Availability

Not applicable.

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
