# Peer review of "Mitochondrial Neurodegeneration: Lessons from Drosophila melanogaster Models"

_biomolecules, 2023, doi:10.3390/biom13020378_

Round 1
Reviewer 1 Report
The authors provide a comprehensive summary of the mitochondrial disease models using the Drosophila melanogaster model system. The review is well written, is organized well and flows nicely. I think that the authors did a comprehensive job describing the results of many different studies in a very digestible and understandable way. I really like the images/Figures that the authors included, especially the schematic of the protein complexes found in the mitochondrial membrane and the mitochondrial genome, which allowed me to connect specific genes with large protein complexes. I only have a couple minor points that I think will help strengthen this manuscript:
1) The images in Figure 2 seem a little pixelated. The authors should include a higher resolution image if possible. Also, the panels in this Figure are not labelled with letters, despite including letters in the Figure legend. Including letters on each of the panels in the image will help to make the Figure easier to follow.
2) I’m not sure that the title of Figure 3 is describing what is being shown in the Figure. The Figure shows all the complexes found in the mitochondrial membrane and the genes that encode for those complexes that have been manipulated to generate Drosophila disease models. The image doesn’t show Drosophila disease models. Updating the figure title might make it more representative of what is being shown in the figure.
3) The text in Tables 1 and 2 are very small. The authors should try to make the text larger so that it is easier to read.
4) I noticed some grammatical errors in the text, so I would recommend the authors proofread the manuscript.
Author Response
We thank the reviewer for his/her appreciation of our work. We changed Figures 2 and 3 to increase readability. We also changed the Title of Figure 3 as requested. We replaced the Tables and provided them as .docx to facilitate editorial manipulation. Finally, we carefully edited the manuscript and corrected grammar errors.
Reviewer 2 Report
Thanks for the excellent review, well-written and with useful graphics. Several short sections could be included in the wrap-up to illustrate increased relevance to the overall series, in particular the usefulness of the fly models for study of disease beyond the classical genetic “mitochondrial diseases.” For example:
•Discussion of the potential use of the fly models to study the overlap of infections disease with mitochondrial metabolism- for example with references with papers such as Drosophila as a Model for Human Viral Neuroinfections - PubMed (nih.gov) Cells 2022 Aug 29;11(17):2685. doi: 10.3390/cells11172685 •Potential to address mechanism environmental factors/ for example Neuroprotective action of 4-Hydroxyisophthalic acid against paraquat-induced motor impairment involves amelioration of mitochondrial damage and neurodegeneration in Drosophila - PubMed (nih.gov) Neurotoxicology 2018 May;66:160-169. doi: 10.1016/j.neuro.2018.04.006. •The potential to study specific cells such as the glia –e.g. Glial α-synuclein promotes neurodegeneration characterized by a distinct transcriptional program in vivo - PubMed (nih.gov) 2019 Oct;67(10):1933-1957. doi: 10.1002/glia.23671. Inflammation in general could shed unique light on pathways involved more generally in disease e.g. Beyond Host Defense: Deregulation of Drosophila Immunity and Age-Dependent Neurodegeneration - PubMed (nih.gov) Front Immunol 2020 Jul 22;11:1574
Author Response
We thank the reviewer for his/her very positive comments. Concerning the suggestion of discussing the use of Drosophila models for other diseases outside primary mitochondrial disorders, we think this falls beyond the scope of this review, which is strictly focused on the use of fly models to unravel the mechanisms of genetically determined mitochondrial diseases.